# Regional Differences in COVID-19 Vaccine Hesitancy in December 2020: A Natural Experiment in the French Working-Age Population

**DOI:** 10.3390/vaccines9111364

**Published:** 2021-11-20

**Authors:** Fanny Velardo, Verity Watson, Pierre Arwidson, François Alla, Stéphane Luchini, Michaël Schwarzinger

**Affiliations:** 1Department of Methodology and Innovation in Prevention, Bordeaux University Hospital, 33000 Bordeaux, France; fanny.velardo@chu-bordeaux.fr (F.V.); francois.alla@u-bordeaux.fr (F.A.); 2University of Bordeaux, Inserm UMR 1219-Bordeaux Population Health, 33000 Bordeaux, France; 3Health Economics Research Unit (HERU), University of Aberdeen, Aberdeen AB25 2ZD, UK; v.watson@abdn.ac.uk; 4Santé Publique France, 94410 Saint Maurice, France; pierre.arwidson@santepubliquefrance.fr; 5CNRS, EHESS, Centrale Marseille, AMSE, Aix-Marseille University, 13001 Marseille, France; stephane.luchini@univ-amu.fr

**Keywords:** SARS-CoV-2, COVID-19, mass vaccination, anti-vaccination behavior, vaccine hesitancy, survey experiment, discrete choice experiment, France

## Abstract

It can be assumed that higher SARS-CoV-2 infection risk is associated with higher COVID-19 vaccination intentions, although evidence is scarce. In this large and representative survey of 6007 adults aged 18–64 years and residing in France, 8.1% (95% CI, 7.5–8.8) reported a prior SARS-CoV-2 infection in December 2020, with regional variations according to an East–West gradient (*p* < 0.0001). In participants without prior SARS-CoV-2 infection, COVID-19 vaccine hesitancy was substantial, including 41.3% (95% CI, 39.8–42.8) outright refusal of COVID-19 vaccination. Taking into account five characteristics of the first approved vaccines (efficacy, duration of immunity, safety, country of the vaccine manufacturer, and place of administration) as well as the initial setting of the mass vaccination campaign in France, COVID-19 vaccine acceptance would reach 43.6% (95% CI, 43.0–44.1) at best among working-age adults without prior SARS-CoV-2 infection. COVID-19 vaccine acceptance was primarily driven by vaccine characteristics, sociodemographic and attitudinal factors. Considering the region of residency as a proxy of the likelihood of getting infected, our study findings do not support the assumption that SARS-CoV-2 infection risk is associated with COVID-19 vaccine acceptance.

## 1. Introduction

On 11 March 2020, the World Health Organization declared the severe acute respiratory syndrome coronavirus 2 (SARS-CoV-2) outbreak in China to be a pandemic [1]. All European countries implemented “stop and go” strategies with more or less restrictive non-pharmaceutical interventions during 2020 to reduce SARS-CoV-2 transmission [2,3] and coronavirus disease-19 (COVID-19) mortality [4,5]. By mid-November 2020, COVID-19 vaccination had become a reality with the announcement of safe and effective candidate vaccines meeting all primary endpoints in large phase 3 trials [6,7,8]. Mass vaccination campaigns started in the European Union (EU) immediately after the approval of a first COVID-19 vaccine on 21 December 2020 [9]. However, beyond global constraints on vaccine supply for mass vaccination campaigns, COVID-19 vaccine hesitancy may represent a major hurdle to reaching herd immunity and controlling the SARS-CoV-2 pandemic without non-pharmaceutical interventions in each country [10,11,12].

Vaccine hesitancy is defined by a delay in acceptance or refusal of vaccination despite availability of vaccination services [13]. Vaccine hesitancy is multifactorial [14] and varies across countries and over time [15]. In this regard, COVID-19 vaccine hesitancy is no exception [16,17,18]. However, high-income countries were diversely affected by the SARS-CoV-2 pandemic [4,5]. It may be assumed that countries with higher prevalence rates of SARS-CoV-2 infection may experience higher COVID-19 vaccination intentions [19,20,21] and reach herd immunity at a faster rate due to the combined effect of higher natural immunity and vaccine uptake.

The large and rapid initial peak of SARS-CoV-2 infection in the North–East and Ile-de-France regions of France [22] provides a unique natural experiment to understand the association between SARS-CoV-2 infection risk and COVID-19 vaccination intentions at the regional level. The main aim of this study was to assess this association in the French working-age population (18–64 years) who is at high risk of SARS-CoV-2 infection as compared to the elderly [22,23]. We conducted a survey experiment in December 2020 among a large and representative sample of twelve regions of France and found that COVID-19 vaccine hesitancy was substantial and primarily driven by vaccine characteristics, sociodemographic and attitudinal factors, but unrelated to SARS-CoV-2 infection risk.

## 2. Materials and Methods

### 2.1. Study Design

We conducted a cross-sectional survey among a representative sample of 6007 adults aged 18–64 years residing in twelve of thirteen regions of France (Auvergne-Rhône-Alpes (ARA), Bourgogne-Franche-Comté (BFC), Bretagne (BRE), Centre-Val de Loire (CVL), Grand Est (GES), Hauts-de-France (HDF), Ile-de-France (IDF), Normandie (NOR), Nouvelle-Aquitaine (NAQ), Occitanie (OCC), Pays-de-la-Loire (PDL), and Provence-Alpes-Côte-d’Azur (PAC)). The survey was fielded from 30 November to 16 December 2020, during the second wave of the SARS-CoV-2 pandemic and the regional extension of the curfew to the whole country [2]. Participants were selected for this survey from an online survey research panel, which was developed and is maintained by BVA (Paris, France) and consists of more than 700,000 French adults. Pre-existing information on the participants was used by BVA to determine eligibility and draw a stratified random sample, with oversampling of participants with low response rates. Sampling was stratified to match French official census statistics by region of residency (gender, age group, education level, household size, and area of residence). The survey was not conducted in Corsica, as this thirteenth region was underrepresented in the BVA online survey research panel.

Participants completed a self-reported online questionnaire eliciting background information, past vaccination behavior, risk factors of a severe form of COVID-19, COVID-19 experience, and risk perceptions about COVID-19 [24]. At the end of the online questionnaire, participants were randomized to two survey experiments assessing COVID-19 vaccine hesitancy: 20% completed the same survey experiment that was fielded in June 2020 [24], and the results of this replication study are reported elsewhere [25]; 80% completed a new survey experiment, and the study results across French regions are reported in this paper (Figure 1).

### 2.2. New Survey Experiment Conducted in December 2020

The survey experiment consisted of two sections [24]: background information on COVID-19 vaccination and the elicitation of vaccine acceptance based on vaccine characteristics. In the first section, all participants were provided with a full page presenting general information about COVID-19 vaccination. In addition, participants were randomized to two different information blocks according to a full factorial design. In block 1, participants were randomized to information about the collective benefits of herd immunity with a herd immunity target in adults aged 18–64 years (>2/3 adults, >1/2 adults, or no information) [26]. In block 2, participants were randomized across the general practitioner’s advice about COVID-19 vaccination (recommendation or no opinion) [27,28].

In the second section, COVID-19 vaccine acceptance was elicited using a discrete choice experiment [29]. Participants completed a series of eight tasks. In each task, they chose between having one of two hypothetical vaccines or no vaccination. As compared to our previous survey experiment [24], the hypothetical vaccines differed on five rather than four characteristics (the duration of immunity conferred by vaccination was added) and four rather than two levels of vaccine safety. The five characteristics (levels) were vaccine efficacy to reduce COVID-19 risk (50%, 80%, 90%, or 100%); vaccine safety regarding the risk serious side-effects (unknown, 1/10,000, 1/100,000, or 1/1,000,000 vaccinated people); country of the vaccine manufacturer (headquarters in EU, United States of America (USA), or China); place of administration (general practitioner practice, local pharmacy, or mass vaccination center); and duration of vaccine immunity (booster needed every 6 months, booster needed every year, or no booster needed).

Based on these vaccine characteristics and levels, there were 432 hypothetical vaccines and 93,096 possible choice tasks between two hypothetical vaccines. A D-efficient experimental design was created with NGENE software (ChoiceMetrics, Sydney, NSW, Australia, 2014) to reduce these to 16 choice tasks with 32 hypothetical vaccines, and participants were randomized to one of two blocks of eight task choices (Appendix A). A choice task example is provided in Figure 2. 

### 2.3. Statistical Analysis

The proportion of participants who reported prior SARS-CoV-2 infection was estimated in the whole sample, overall and by region, with use of an exact 95% confidence interval (95% CI) for a binomial proportion.

COVID-19 vaccine acceptance was estimated in the sample without prior SARS-CoV-2 infection with use of a single hurdle repeated discrete choice model [24,30]. Briefly, this behavioral model is a two-part model that specifies one decision process underlying outright vaccination refusal (i.e., serial non-vaccination in all eight choice tasks) and another decision process underlying vaccine acceptance (or conversely vaccine hesitancy) depending on vaccine characteristics. The probability of outright vaccination refusal and associated independent factors was analyzed using a logit regression model. The probability of vaccine acceptance based on vaccine characteristics and other independent factors was analyzed using a conditional logit regression model for repeated discrete choices in participants who did not refuse vaccination outright. Altogether, the two-part model provides an estimation of COVID-19 vaccine acceptance, i.e., (1 − p_outright vaccination refusal_) × p_vaccine acceptance_.

The same set of independent factors was used to explain both decision processes and included the following: region of residency, six stratification variables used in the sampling procedure by region (gender, age group, educational level, household size (number of adults and number of children), and area of residence); 15 variables related to vaccination behavior (working status including health-care worker status, unemployment or furlough because of the COVID-19 health crisis, compliance with recommended vaccinations in the past, vaccination against 2020/21 influenza, six risk factors for a severe form of COVID-19 (pregnancy, smoking status, obesity, hypertension, diabetes, and other chronic condition), four variables related to COVID-19 experience (COVID-19 symptoms, test for SARS-CoV-2 infection diagnosis, knowledge of people who got COVID-19, and curfew in the municipality), and perceived severity of COVID-19 if infected); and the randomized information blocks.

Probabilities and 95% CI of outright vaccination refusal and COVID-19 vaccine acceptance were estimated at the sample mean, overall and by region, for the first approved vaccines [9]. Characteristics of the first approved vaccines differed on efficacy (Pfizer&BioNTech or Moderna: 95% vs. AstraZeneca: 60%) [9,31,32,33], safety (Pfizer&BioNTech or Moderna: 1/1,000,000 vs. AstraZeneca: 1/100,000) [9,34], country of the vaccine manufacturer (Pfizer or Moderna: USA vs. BioNTech or AstraZeneca: EU) [9], and place of administration in the initial setting of the mass vaccination campaign in France (Pfizer&BioNTech or Moderna: mass vaccination center vs. AstraZeneca: general practitioner practice or local pharmacy). Other vaccination characteristics were controlled for at the same level (low duration of vaccine immunity with a booster needed every 6 months; vaccination recommended by the general practitioner; and no information provided about the collective benefits of herd immunity in the initial setting of the mass vaccination campaign in France).

All analyses were based on two-sided *p*-values, with *p* < 0.05 considered to indicate statistical significance. The single hurdle repeated discrete choice model was estimated using maximum likelihood techniques with R statistical software [24].

## 3. Results

### 3.1. Prevalence of Detected SARS-CoV-2 Infection in the French Working-Age Population

A representative sample of 6007 adults aged 18–64 years residing in twelve regions of France participated in the online survey from 30 November to 16 December 2020. Of them, 8.1% (95% CI, 7.5–8.8) reported a prior SARS-CoV-2 infection (1.1% hospital admission for COVID-19; 1.7% medical diagnosis of COVID-19 without a virology test; and 5.3% positive test for SARS-CoV-2 infection diagnosis with (2.9%) or without (2.4%) symptoms). The prevalence rate of detected SARS-CoV-2 infection significantly varied across regions (*p* < 0.0001) according to an East–West gradient (Figure 3; Appendix A).

### 3.2. COVID-19 Vaccine Hesitancy in Participants without Prior SARS-CoV-2 Infection

A total of 4806 (80%) participants were randomly allocated to the new survey experiment assessing COVID-19 vaccine hesitancy depending on five vaccine characteristics (Figure 1). Of them, 4415 (91.9%) did not report prior SARS-CoV-2 infection and were selected for the analysis of COVID-19 vaccine hesitancy. As expected from differences between regional populations, sociodemographic characteristics of the selected participants significantly varied across regions (age: *p* < 0.05; education: *p* < 0.001; and area of residence: *p* < 0.001) (Appendix A). In addition, participants residing in regions with higher prevalence rates of detected SARS-CoV-2 infection more frequently reported COVID-19 experience (having had COVID-19 symptoms without medical confirmation: *p* < 0.001; and knowing someone who got COVID-19: *p* < 0.001).

A total of 1823 (41.3% (95% CI, 39.8–42.8)) participants were identified as refusing COVID-19 vaccination outright. Outright vaccination refusal was not associated with the region of residency in univariate (overall test: *p* = 0.18; Appendix A) and multivariate analyses (overall test: *p* = 0.60; Table 1). By contrast, it was independently associated with female gender (*p* < 0.0001), age with an inverted U-shaped relationship (*p* < 0.0001), lower educational achievement (*p* < 0.01), poorer compliance with recommended vaccinations in the past (*p* < 0.0001), no vaccination against 2020/21 influenza (*p* < 0.0001), and lower perceived severity of COVID-19 if infected (*p* < 0.0001).

Among 2592 (58.7%) participants who did not refuse vaccination outright, vaccine hesitancy was associated with the region of residency (overall test: *p* < 0.01; Table 1) due to two outlier regions with higher vaccine hesitancy: Bourgogne-Franche-Comté (BFC) (*p* < 0.05) and Occitanie (OCC) (*p* < 0.01). By contrast, vaccine hesitancy was primarily decreased with higher vaccine efficacy, lower risk of serious side-effects, vaccines made in EU rather than USA (or China), and, to a lesser extent, longer duration of vaccine immunity (all *p* < 0.0001). In addition, sociodemographic and attitudinal factors associated with outright vaccination refusal were also strongly associated with vaccine hesitancy, albeit strengths of association were generally weaker with vaccine hesitancy.

### 3.3. COVID-19 Vaccine Acceptance Predicted in December 2020

In the initial setting of the mass vaccination campaign in France, the behavioral model predicted that outright refusal of COVID-19 vaccination would be present in 42.1% (95% CI, 41.6–42.7) of the French working-age population without prior SARS-CoV-2 infection (Figure 4). According to the characteristics of the first approved vaccines, COVID-19 vaccine acceptance would reach at best 43.6% (95% CI, 43.0–44.1) for a BioNTech vaccine promoted as made in EU rather than USA (Figure 4) and would be lower for Pfizer or Moderna vaccines (38.4% (95% CI, 37.8–38.9)); Appendix A) and the AstraZeneca vaccine (31.1% (95% CI, 30.6–31.6)); Appendix A).

Acceptance of a BioNTech vaccine was estimated at the sample mean of each region of residency. Figure 4 presents regional estimates ordered by increasing prevalence of detected SARS-CoV-2 infection. Outright refusal of COVID-19 vaccination would significantly decrease from 48.0% (95% CI, 46.1–49.9) in Occitanie (OCC) to 37.9% (95% CI, 35.7–40.1) in Pays-de-la-Loire (PDL), although it was not correlated with the prevalence of detected SARS-CoV-2 infection (ρ = 0.27; *p* = 0.40). Similarly, acceptance of a BioNTech vaccine would significantly increase from 37.2% (95% CI, 35.4–39.0) in Occitanie (OCC) to 48.9% (95% CI, 46.5–51.3) in Bretagne (BRE), although it was not correlated with the prevalence of detected SARS-CoV-2 infection (ρ = −0.26; *p* = 0.42).

## 4. Discussion

In this large and representative survey of 6007 adults aged 18–64 years and residing in France, 8.1% (95% CI, 7.5–8.8) reported a prior SARS-CoV-2 infection in December 2020, with regional variations according to an East–West gradient (*p* < 0.0001). In participants without prior SARS-CoV-2 infection, COVID-19 vaccine hesitancy was substantial, including 41.3% (95% CI, 39.8–42.8) outright refusal of COVID-19 vaccination. Taking into account five characteristics of the first approved vaccines (efficacy, duration of immunity, safety, country of the vaccine manufacturer, and place of administration) as well as the initial setting of the mass vaccination campaign in France, COVID-19 vaccine acceptance would reach 43.6% (95% CI, 43.0–44.1) at best among working-age adults without prior SARS-CoV-2 infection. COVID-19 vaccine acceptance was primarily driven by vaccine characteristics, sociodemographic and attitudinal factors. Considering the region of residency as a proxy of the likelihood of getting infected, our study findings do not support the assumption that SARS-CoV-2 infection risk is associated with COVID-19 vaccine acceptance.

To our knowledge, this is the first study assessing the association between SARS-CoV-2 infection risk and COVID-19 vaccination intentions. France provides a natural experiment to assess this association as SARS-CoV-2 pandemic initially spread in the North–East and Ile-de-France regions and was controlled by a nationwide lockdown ordered on 17 March 2020 [2]. Accordingly, the heterogeneity of SARS-CoV-2 infection rates across regions was high after the first wave (11 May 2020) and only marginally reduced during the second wave (31 October 2020) [22]. In our study, 8.1% (95% CI, 7.5–8.8) of participants reported a prior SARS-CoV-2 infection during the second wave of the SARS-CoV-2 pandemic in December 2020. In comparison, the detection rate of SARS-CoV-2 infection in the French working-age population was similarly estimated at 7.8% (95% CI, 6.0–10.0) as of 15 January 2021 in an epidemiological modeling study [22]. In addition, the external validity of our study was enhanced by the significant correlation of regional estimates between the two studies (ρ = 0.81; *p* < 0.01) (Appendix A).

We assessed COVID-19 vaccination intentions in participants without prior SARS-CoV-2 infection with use of a two-part behavioral model [24]. As compared to opinion surveys asking about vaccination intentions for vaccines with unspecified characteristics [16,17,18], it allowed the identification of participants who would never choose to be vaccinated regardless of five vaccine characteristics (outright refusal of vaccination) and those who are hesitant and whose choice to be vaccinated would depend on vaccine characteristics [9]. In participants who did not refuse vaccination outright, we found that vaccine efficacy, vaccine safety, and the country of the vaccine manufacturer were primary drivers of COVID-19 vaccine acceptance as in previous discrete choice experiments [24,25,35,36,37,38]. In addition, participants were randomly allocated to two discrete choice experiment designs presenting five (80%) vs. four (20%) vaccine characteristics (Figure 1), and we found robust estimates across the two survey experiments regarding outright refusal of vaccination (41.3% (95% CI, 39.8–42.8)) vs. 42.5% (95% CI, 41.1–43.8), respectively) and overall acceptance of the best (BioNTech) vaccine (43.6% (95% CI, 43.0–44.1) vs. 44.0% (95% CI, 42.7–45.2), respectively) [25].

Our study findings do not support the assumption that SARS-CoV-2 infection risk is associated with COVID-19 vaccination intentions in the working-age population. At the individual level, the region of residency was not an independent predictor of outright vaccination refusal. Two regions (Bourgogne-Franche-Comté and Occitanie) were independently associated with higher vaccine hesitancy as compared to other regions. However, they were in the middle range rather than the ends of prevalence rates of detected SARS-CoV-2 infection (Appendix A). At the macro level, COVID-19 vaccine acceptance significantly varied across regions, although it was uncorrelated with prevalence rates of detected SARS-CoV-2 infection (ρ = −0.26; *p* = 0.42). By contrast, regional variations in COVID-19 vaccine acceptance were primarily explained by differences on sociodemographic characteristics (age, education, and area of residency) that were in turn consistently associated with COVID-19 vaccine hesitancy [24,25].

The main limitation of our study relates to its timing. COVID-19 vaccination intentions may continuously evolve in relation to the dynamic of the SARS-CoV-2 pandemic and associated non-pharmaceutical interventions [2,3] as well as information on COVID-19 vaccine uptake made available to the public in almost real-time [39]. Nevertheless, our study findings support that COVID-19 vaccine hesitancy was profoundly anchored in multiple, large sociodemographic strata of the working-age population (women, 25–54 years old, lower educational achievement, rural areas) before mass vaccination started in the elderly in France.

Another limitation is that sample size was limited in regions that are sparsely populated, with a minimum of 234 (3.9%) participants in the Centre-Val de Loire (CVL) region. However, the study was conducted among the working-age population who is at high risk of SARS-CoV-2 infection as compared to the elderly [22,23]. To avoid spurious association, we excluded participants reporting prior SARS-CoV-2 infection from the analysis of COVID-19 vaccine hesitancy as they were excluded from vaccine phase 3 trials [31,32,33], and vaccination was not recommended for them in the initial setting of the mass vaccination campaign in France [40]. In addition, our behavioral model of COVID-19 vaccine hesitancy was controlled for a large set of independent factors including COVID-19 experience and the perceived severity of COVID-19 that may confound the association between the likelihood of getting infected and COVID-19 vaccination intentions [19,20,21].

Finally, Corsica was not included in our study as this region was underrepresented in the BVA online survey research panel. Corsica accounts for 0.5% of the working-age population in France (2020 national census data), and its inclusion may have biased our correlation estimate between the prevalence of detected SARS-CoV-2 infection and COVID-19 vaccination hesitancy at the regional level. In addition, Corsica was associated with one of the lowest detection rates of SARS-CoV-2 infection as of 15 January 2021 (4.0% (95% CI, 3.1–5.3) of the working-age population) [22] but remained in the middle range of COVID-19 vaccination coverage recorded by region (e.g., 42.2% as compared to 43.4% (min: 40.0%; max: 47.0%) of the working-age population in other regions of metropolitan France as of 11 July 2021) [41], further supporting that there is no association between SARS-CoV-2 infection risk and COVID-19 vaccination in the working-age population.

In summary, our survey experiment supports that COVID-19 vaccine hesitancy was deeply anchored in the French working-age population in December 2020 and primarily driven by vaccine characteristics, sociodemographic and attitudinal factors, but unrelated to SARS-CoV-2 infection risk at the rather low levels estimated in 2020. Accordingly, French regions that were initially spared during the first wave of SARS-CoV-2 pandemic may remain at high risk of local outbreaks of SARS-CoV-2 pandemic despite availability of COVID-19 vaccination services [10,11,12].

## Figures and Tables

**Figure 1 vaccines-09-01364-f001:**
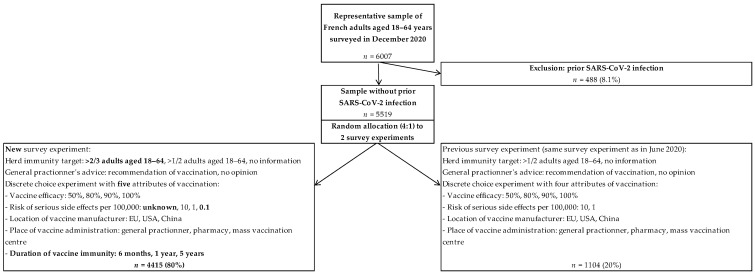
Study flowchart.

**Figure 2 vaccines-09-01364-f002:**
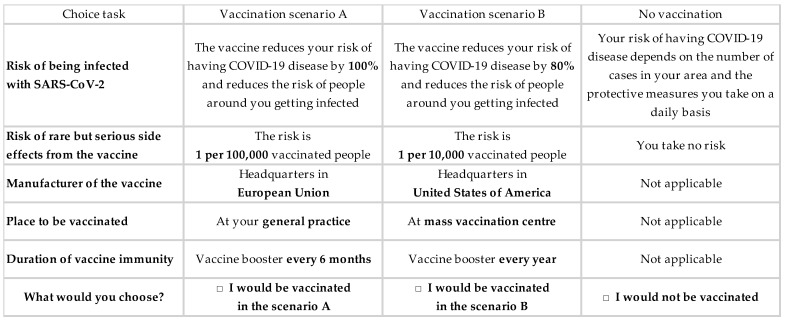
Choice task example in the discrete choice experiment used in December 2020.

**Figure 3 vaccines-09-01364-f003:**
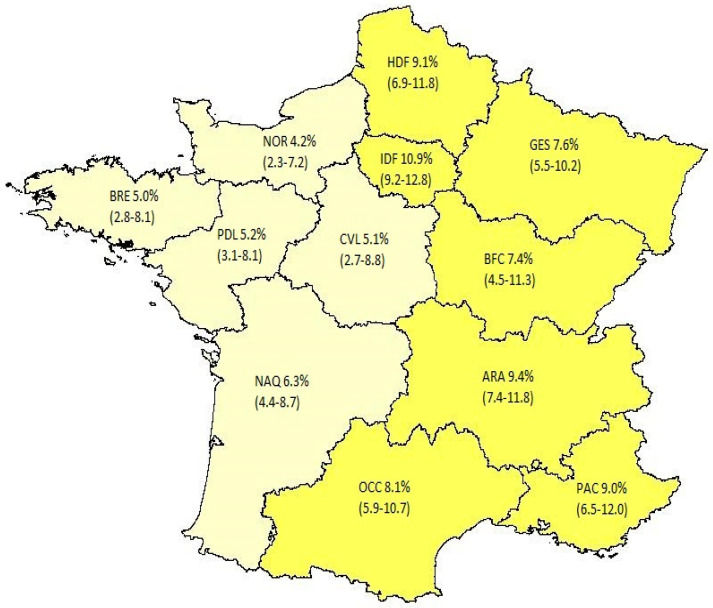
Prevalence of detected SARS-CoV-2 infection in the French working-age population (December 2020). Figure legend: Auvergne-Rhône-Alpes (ARA), Bourgogne-Franche-Comté (BFC), Bretagne (BRE), Cen-tre-Val de Loire (CVL), Grand Est (GES), Hauts-de-France (HDF), Ile-de-France (IDF), Normandie (NOR), Nouvelle-Aquitaine (NAQ), Occitanie (OCC), Pays-de-la-Loire (PDL), and Provences-Alpes-Côte-d’Azur (PAC).

**Figure 4 vaccines-09-01364-f004:**
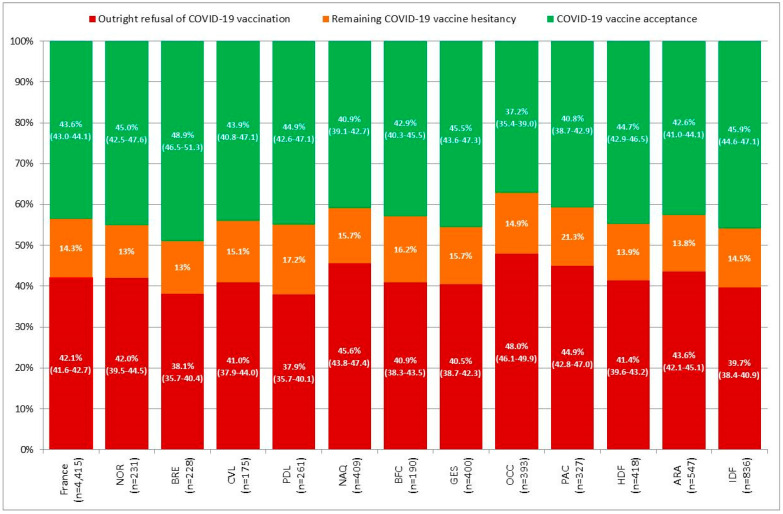
Acceptance of BioNTech vaccine predicted in the French working-age population, overall and by region (December 2020). Figure legend: Auvergne-Rhône-Alpes (ARA), Bourgogne-Franche-Comté (BFC), Bretagne (BRE), Cen-tre-Val de Loire (CVL), Grand Est (GES), Hauts-de-France (HDF), Ile-de-France (IDF), Normandie (NOR), Nouvelle-Aquitaine (NAQ), Occitanie (OCC), Pays-de-la-Loire (PDL), and Provences-Alpes-Côte-d’Azur (PAC). French regions of residency are ordered by detected SARS-CoV-2 prevalence.

**Table 1 vaccines-09-01364-t001:** Behavioral model of COVID-19 vaccination in the French working-age population without prior SARS-CoV-2 infection in December 2020 (*n* = 4415).

		Outright Refusal of COVID-19 Vaccination (41.3%)		COVID-19 Vaccine Hesitancy (58.7%)	
Determinants of COVID-19 Vaccination		OR (95% CI)	*p*-Value	OR (95% CI)	*p*-Value
French region of residency (ordered by detected SARS-CoV-2 prevalence)					
	Normandie (NOR) (ref. Ile-de-France)	0.92 (0.65–1.31)	0.66	0.92 (0.8–1.10)	0.36
	Bretagne (BRE) (ref. Ile-de-France)	0.79 (0.53–1.16)	0.23	0.84 (0.73–1.00)	0.079
	Centre-Val de Loire (CVL) (ref. Ile-de-France)	0.81 (0.53–1.23)	0.32	1.09 (0.89–1.39)	0.42
	Pays-de-la-Loire (PDL) (ref. Ile-de-France)	0.73 (0.49–1.06)	0.10	1.12 (0.93–1.41)	0.21
	Nouvelle-Aquitaine (NAQ) (ref. Ile-de-France)	1.05 (0.74–1.49)	0.79	1.02 (0.86–1.23)	0.85
	Bourgogne-Franche-Comté (BFC) (ref. Ile-de-France)	0.92 (0.61–1.38)	0.69	1.22 (1.00–1.61)	0.041
	Grand Est (GES) (ref. Ile-de-France)	0.88 (0.63–1.23)	0.47	0.97 (0.84–1.16)	0.75
	Occitanie (OCC) (ref. Ile-de-France)	1.10 (0.83–1.47)	0.50	1.23 (1.05–1.49)	0.004
	Provence-Alpes-Côte-d’Azur (PAC) (ref. Ile-de-France)	0.96 (0.70–1.32)	0.82	1.02 (0.89–1.21)	0.76
	Hauts-de-France (HDF) (ref. Ile-de-France)	0.97 (0.73–1.30)	0.85	0.99 (0.87–1.15)	0.91
	Auvergne-Rhône-Alpes (ARA) (ref. Ile-de-France)	1.05 (0.81–1.36)	0.72	1.01 (0.90–1.16)	0.88
Vaccine characteristics					
	Vaccine efficacy of 80% (ref. 50%)	Not applicable		0.58 (0.56–0.60)	<0.0001
	Vaccine efficacy of 90% (ref. 50%)			0.44 (0.43–0.45)	<0.0001
	Vaccine efficacy of 100% (ref. 50%)			0.33 (0.32–0.34)	<0.0001
	Vaccine booster every year (ref. booster every 6 months)			0.95 (0.90–0.99)	0.029
	No vaccine booster needed (ref. booster every 6 months)			0.78 (0.75–0.81)	<0.0001
	Unknown risk of serious side effects (ref. 1/10,000)			1.54 (1.40–1.71)	<0.0001
	Risk of serious side effects of 1/100,000 (ref. 1/10,000)			0.76 (0.73–0.80)	<0.0001
	Risk of serious side effects of 1/1,000,000 (ref. 1/10,000)			0.57 (0.55–0.59)	<0.0001
	Vaccine made in USA (ref. European Union)			1.59 (1.44–1.77)	<0.0001
	Vaccine made in China (ref. European Union)			2.40 (2.13–2.74)	<0.0001
	Vaccination at GP practice (ref. mass vaccination center)			0.98 (0.93–1.03)	0.42
	Vaccination at local pharmacy (ref. mass vaccination center)			0.98 (0.93–1.03)	0.37
Background information on COVID-19 vaccination					
	>67% of adults aged 18–64 years old must be immunized to reach herd immunity (ref. no information)	0.89 (0.76–1.04)	0.16	0.92 (0.86–0.99)	0.029
	>50% of adults aged 18–64 years old must be immunized to reach herd immunity (ref. no information)	0.99 (0.84–1.16)	0.88	1.05 (0.97–1.14)	0.22
	General practitioner recommends vaccination (ref. no opinion)	1.01 (0.88–1.15)	0.90	1.02 (0.96–1.09)	0.52
Survey stratification variables (by region of residency)					
	Men (ref. women)	0.55 (0.48–0.64)	<0.0001	0.71 (0.67–0.74)	<0.0001
	25–34 years (ref. 18–24)	1.60 (1.21–2.12)	0.001	1.02 (0.90–1.18)	0.81
	35–44 years (ref. 18–24)	1.70 (1.28–2.26)	0.0003	1.04 (0.91–1.20)	0.61
	45–54 years (ref. 18–24)	1.69 (1.28–2.24)	0.0002	0.89 (0.79–1.00)	0.08
	55–64 years (ref. 18–24)	1.48 (1.11–1.99)	0.008	0.86 (0.77–0.98)	0.04
	High school graduate (ref. some high school)	0.89 (0.75–1.06)	0.20	0.86 (0.80–0.93)	0.001
	University graduate (ref. some high school)	0.75 (0.62–0.90)	0.002	0.88 (0.82–0.96)	0.007
	Number of adults in the household	0.98 (0.88–1.09)	0.74	1.11 (1.05–1.18)	<0.0001
	Number of children in the household	1.02 (0.94–1.11)	0.57	1.01 (0.98–1.06)	0.48
	Urban area <100,000 inhabitants (ref. rural area)	0.94 (0.78–1.13)	0.53	1.00 (0.92–1.10)	0.97
	Urban area ≥100,000 inhabitants (ref. rural area)	0.78 (0.64–0.94)	0.010	0.98 (0.90–1.08)	0.66
Variables related to vaccination behavior					
	Healthcare worker (ref. not working)	1.01 (0.78–1.31)	0.93	1.25 (1.08–1.49)	0.0005
	Worker in contact with the public (ref. not working)	0.95 (0.78–1.14)	0.56	1.14 (1.03–1.27)	0.005
	Worker not in contact with the public (ref. not working)	1.08 (0.89–1.3)	0.45	1.14 (1.03–1.27)	0.009
	Unemployed or furloughed because of the COVID-19 health crisis (ref. not reported)	0.97 (0.81–1.16)	0.76	0.95 (0.87–1.03)	0.20
	Sometimes compliant with recommended vaccination in the past (ref. always)	2.15 (1.86–2.48)	<0.0001	1.61 (1.45–1.82)	<0.0001
	Never compliant with recommended vaccination in the past (ref. always)	5.21 (4.22–6.44)	<0.0001	2.92 (2.16–4.51)	<0.0001
	Vaccination against influenza 2020/2021 (ref. not reported)	0.54 (0.42–0.67)	<0.0001	0.61 (0.58–0.65)	<0.0001
	Pregnancy (ref. not reported)	1.46 (0.73–2.97)	0.28	1.37 (0.91–2.82)	0.10
	Current smoker (ref. never smoker)	0.93 (0.79–1.10)	0.40	1.04 (0.95–1.13)	0.41
	Former smoker (ref. never smoker)	0.96 (0.82–1.12)	0.62	1.13 (1.04–1.24)	0.002
	Obese (ref. normal weight)	1.04 (0.86–1.26)	0.68	1.05 (0.96–1.16)	0.29
	Overweight (ref. normal weight)	1.15 (0.99–1.34)	0.07	1.14 (1.05–1.24)	0.0006
	Hypertension (ref. not reported)	0.77 (0.59–1.00)	0.050	0.84 (0.76–0.93)	0.004
	Diabetes (ref. not reported)	0.80 (0.55–1.15)	0.23	1.09 (0.93–1.32)	0.31
	Other chronic condition (ref. not reported)	0.99 (0.76–1.28)	0.92	1.09 (0.96–1.25)	0.17
	Had COVID-19 symptoms (ref. not reported)	0.95 (0.79–1.13)	0.54	1.10 (1.00–1.21)	0.031
	Had a test for SARS-CoV-2 infection diagnosis (ref. not reported)	1.01 (0.84–1.20)	0.94	0.83 (0.78–0.90)	<0.0001
	Knows someone who got COVID-19 with hospital admission (ref. not reported)	0.81 (0.67–0.98)	0.032	1.12 (1.01–1.25)	0.018
	Knows someone who got COVID-19 without hospital admission (ref. not reported)	0.9 (0.77–1.04)	0.15	1.07 (1.00–1.17)	0.06
	Curfew in the municipality after 17 October 2020 (ref. after 15 December 2020)	0.94 (0.73–1.23)	0.67	0.92 (0.82–1.04)	0.21
	Curfew in the municipality after 28 October 2020 (ref. after 15 December 2020)	0.94 (0.78–1.14)	0.53	0.87 (0.81–0.95)	0.005
Perceived severity of COVID-19 if infected					
	Very severe (ref. Not severe at all)	0.46 (0.32–0.65)	<0.0001	0.72 (0.64–0.82)	0.0001
	Somewhat severe (ref. Not severe at all)	0.55 (0.43–0.71)	<0.0001	0.65 (0.60–0.71)	<0.0001
	Not particularly severe (ref. Not severe at all)	0.74 (0.58–0.94)	0.015	0.75 (0.69–0.83)	<0.0001
	Don’t know (ref. Not severe at all)	0.82 (0.63–1.08)	0.16	1.09 (0.94–1.28)	0.24

## Data Availability

De-identified cross-sectional data and R program used in this study can be made available on reasonable request from the corresponding author.

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
