# Peer review of "Regional Differences in COVID-19 Vaccine Hesitancy in December 2020: A Natural Experiment in the French Working-Age Population"

_vaccines, 2021, doi:10.3390/vaccines9111364_

Round 1

Reviewer 1 Report

Estimated Authors,

I've read with great interest the present paper entitled: "COVID-19 Vaccine Hesitancy in the French Working Age Population: (Replication Experiment in December 2020)". The present research from the study group lead by Michaël Schwarzinger reports on the VH in a sample drawn from the general population of France, discussing both the occurrence of VH and its main drivers during a specific time period (i.e. shortly before the inception of the vaccination campaign in HCWs).

The results from this study, albeit not radically new, may be of certain interest for the international readers. Unfortunately, in my opinion, the present paper cannot be accepted by VACCINES for the following reasons:

1) the paper is a follow up of the original study (doi: 10.1016/S2468-2667(21)00012-8) that was published on Lancet Public Health some month ago. The sample was quite different (i.e. 2000 participants vs. 6000), and the time frame as well. Therefore, this is not truly a problem. On the contrary, the present paper shares too much similarities (at least in my opinion) with the paper COVID-19 vaccine hesitancy in the French working age popu-2 lation: a replication experiment in December 2020 having Dr. Schwarzinger as First Authors and corresponding Author.

Not only the introduction has quite the same text, but also the following sections share similar pros and cons, including problems in the methodological approach and data reporting. Obviously, the latter paper represents a further follow up, with some differencies in the data reporting (see Table 1: the present research has a more detailed data reporting, particularly when dealing with geographical characteristics of the participants) but I had significant difficulties in catching the true differences in the results and in the key messages of these papers. I warmly recommend that the two paper would be merged in order to remove such main issue.

2) While that from (1) represents a critical issue, the following ones are of methodological nature, and therefore the very next lines include only my recommendations to overcome potential limits in the study text.

a) Authors state that "Participants were selected from the BVA online survey research panel with stratification (gender; age group; education level; household size; and area of residence) and weighted random sampling by region of residency)". Not only the sentence is duplicated from the study of Dr. Schwarzinger ("Participants were selected from the BVA online survey research panel with stratification (gender; age group; education level; household size; and area of residence) and weighted random sampling by region of residency"), but what is "BVA"? In the original research from LANCET we understand that "Participants  were  selected  from   an   online   survey   research   panel,   which   was   developed  and  is  maintained  by  the  opinion  survey  research  firm  BVA  (Paris,  France)  and  consists  of  more  than  700  000  French  adults." However, neither the original study from Lancet nor the present one or the other research paper from Velardo et al explain how the participants were selected by BVA, i.e. for the specific aim of performing this study, or as a referent population for other large scale questionnaire studies. Please explain. Please also explain why did the first report include around 2000 participants, compared to > 6000 in these newer reports. Was it a real-world issue (e.g. the Authors were able to recruit and report on 2000 individuals in the first steps of the analyses, compared to over 6000 in the follow up studies?)? Please also explain whether the original 2000 participants were included in the final specimen of 6000 participants, of were otherwise excluded.

b) Authors performed their analyses "single hurdle repeated discrete choice model was estimated using maximum likelihood techniques with R statistical software". This is the very same model performed in the other aforementioned studies, but some results are quite different, particularly in the study having Dr. Schwarzinger as first Author. For example: 

Dr. Schwarzinger: Table 1, "perceived severity of COVID-19 if infected" --> very severe: OR 0.30 (95%CI 0.15-0.62) / OR 0.78 (95%CI 0.60-1.11)

present study: Table 1, "perceived severity of COVID-19 if infected" --> very severe: 0.46 (95%CI 0.32-0.65) / OR 0.72 (95%CI 0.64-0.82). 

Please explain in full details how the aforementioned estimates may be slightly but considerably different. A detailed explanation may contribute to explain the reasons that motivated your study group to report in two different paper about your study: at the moment, the text is inadequate to discuss such specific issue.

c) in discussion section I would expect some further discussion on the actual VH experienced during the vaccination campaigns. Some hints are reported, particularly when dealing at regional level, but being the vaccination campaign a radically and continously evolving issue, I warmly recommend the authors to more extensively discuss their estimates with the actual figures.

Reviewer 2 Report

Dear Authors,

congratulations for the very good work produced. 

Your paper is of high interest in the current panorama related to COVID-19 infections and vaccines and in particular understanding the factors linked to a higher or lower willingness to vaccinate. 

Tha sample is representative also by a geographical viewpoint and the results and discussion are widely explored. 

I have just one point to be cleared. You wrote that "Ethical review and approval were waived for this study as the survey experiment followed all requirements under French regulations". I would like to have more details about this because, as I know different rules also for observational studies like yours (i.e. ethical approval is always needed) and without ethical approval the study is not suitable for publication. 

Author Response

We thank the reviewer for his/her interest in our study and overall positive assessment of the manuscript.

We agree with the reviewer that requirements for conducting research on human subjects may surprisingly differ across countries. The approval of an ethics committee is the rule for any biomedical research (including France) and was expanded to social sciences research in several countries over the years (e.g., US in 1981; Canada in 1998; UK in 2005). By contrast, the approval of an ethics committee is still not required for a non-interventional research in France (the latest version of the related law in the French Public Health Code is available at: https://www.legifrance.gouv.fr/codes/section_lc/LEGITEXT000006072665/LEGISCTA000006170998/#LEGISCTA000032722874 ).

Nevertheless, our study complies with other requirements of research under French regulations (as in other countries). Participants were selected from the BVA online survey research panel and completed the questionnaire on the BVA online survey platform. In this regard, the data management of BVA preserves the anonymity of participants and is approved by the French National Commission for Data Protection (CNIL) in accordance with the General Data Protection Regulation of the European Union (approval renewal on May 25, 2018). In addition, participants to our online survey were initially informed on the study purpose and gave their informed consent to this research by starting the questionnaire.

Following the reviewer’s comment, we have clarified the “Institutional Review Board Statement” with the following: “The approval of an ethics committee was not required for this non-interventional research study (Act L1121 of French Public Health Code). The data management of the BVA online survey research panel preserves the anonymity of participants and is approved by the French National Commission for Data Protection (CNIL) in accordance with the General Data Protection Regulation of the European Union (approval renewal on May 25, 2018)."

Reviewer 3 Report

In this paper, Velardo et al carried out a large and representative survey on COVID-19 vaccine hesitancy in 6007 adults aged 18-64 years and residing in France in December 2020. They found that COVID-19 vaccine hesitancy was substantial and primarily driven by vaccine characteristics, sociodemographic and attitudinal factors, but unrelated to SARS-CoV-2 infection risk. Overall, the study is interesting, timely, useful, and well-presented.

Strengths: Interesting and timely study carried out in France where the evolution of the outbreak was quite original. Moreover, the paper is well-presented and well-written.

Weaknesses: The study was carried out on December 2020 and the situation in France has changed. Moreover, sample size was low in some Regions. However, authors were well-aware about these weaknesses and honest in their discussion.

Major

-

Moderate

-L68: The twelve regions could be listed here.

-L73: Corsica was not included in the study. What could be anticipated for this region based on the last observations? Any idea. That point could be mentioned in the discussion. At least a reminder stipulating that one Region was not considered.

-Figure 3 could be improved. Indeed, black font in the bars is uneasy to read. Please replace by white font and adapt the text to the size of the bars.

-Figure 3: Abbreviations should be explained in the figure legend.

-L299-305: I suppose “if infection risk had been a lot higher maybe the observations would have been quite different. Here the risk was quite low finally especially considering the strict lockdown. That point could be discussed too.

Minor

-BVA as all the abbreviations should be defined.

-L183: Add a space after 1,823.

-L284: Please replace 25-54 years old by 25-54-year-old.

-The dot after Lancet could be removed for references 12, 22…

Round 2

Reviewer 1 Report

Estimated Authors of the paper "Regional differences in COVID-19 vaccine hesitancy in December 2020: a natural experiment in the French working age 3 population" (it was: "COVID-19 Vaccine Hesitancy Across French Regions: (Natural Experiment in December 2020"),

I've appreciated the considerable efforts you paid in order to cope with my previous concerns and recommendations. 

I must admit that, in the present form, most of my concerns have been removed. Now, the study has a distinctive status, both when compared to the study originally published on Lancet, and when compared to its sister study.

Most of the original concerns were addressed by the Study Authors in the rebuttal letter, whose content I've appreciated. In other words, I'm shifting my original recommendation from "reject" to its substantial acceptance.